# Anatomical Insights into the Lateral Meniscus and Anterolateral Ligament: A Cadaveric Study

**DOI:** 10.3390/diagnostics15232987

**Published:** 2025-11-24

**Authors:** João Lobo, Joana Almeida, José Fernandes, Hélio Alves, André Rodrigues Pinho, Maria Dulce Madeira, Levi Reina Fernandes, Ana Esteves, Pedro Alberto Pereira

**Affiliations:** 1Department of Surgery and Physiology, Faculty of Medicine, University of Porto, Alameda Professor Hernâni Monteiro, 4200-319 Porto, Portugal; 2Orthopedics and Traumatology Department, Unidade Local Saude São João, Alameda Professor Hernâni Monteiro, 4200-319 Porto, Portugal; joana17almeida17@gmail.com (J.A.); jdacunhafernandes@gmail.com (J.F.); arpcinco@hotmail.com (A.R.P.); 3Unit of Anatomy, Department of Biomedicine, Faculty of Medicine, University of Porto, Alameda Professor Hernâni Monteiro, 4200-319 Porto, Portugal; helioalves@med.up.pt (H.A.); madeira@med.up.pt (M.D.M.); pedroper@med.up.pt (P.A.P.); 4NeuroGen Research Group, Center for Health Technology and Services Research (CINTESIS), Rua Dr. Plácido da Costa, 4200-450 Porto, Portugal; 5CINTESIS@RISE, Faculty of Medicine, University of Porto, Alameda Professor Hernâni Monteiro, 4200-319 Porto, Portugal; 6Hospital CUF Santarém, Rua Zeferino Silva 39 a 51, 2005-321 Santarém, Portugal; levirfs@gmail.com; 7Orthopedics and Traumatology Department, Unidade Local Saude do Tâmega e Sousa, Avenida do Hospital Padre Américo, No. 210, 4560-136 Guilhufe, Portugal; ana_silva_esteves@hotmail.com

**Keywords:** anatomical research, anterolateral ligament, cadaveric study, human anatomical dissection, knee, lateral meniscus, surgical anatomy

## Abstract

**Background/Objectives:** This study aims to describe in detail the previously reported close relationship between the anterolateral ligament (ALL) and the lateral meniscus. Few previous studies identified and characterized this relation. This study further characterizes the anatomical relation between the ALL and the lateral meniscus through meticulous cadaveric dissection. **Methods:** A total of 31 cadaver knees were dissected. The ALL relation with the lateral meniscus was explored using a specific dissection protocol that involved removing the central pivot (cruciate ligaments) and medial structures of the knee to enhance visualization of the anterolateral complex. The zone and attachment pattern of the ALL on the lateral meniscus were recorded. **Results:** The ALL was found in all 31 dissected knees and in all cases has an attachment to the lateral meniscus. It was attached in zone 2b of the lateral meniscus in 97% of cases. The median anteroposterior length of attachment of the ALL on the lateral meniscus was 6 mm (25th and 75th percentiles of 5–7 mm). Almost 80% (77.4%) of ALL attachments on the lateral meniscus were full thickness or bipolar (superior and inferior margins). In the remaining knees, the ALL was fixed only in the upper part (4 knees, 12.9%) or only in the lower part (3 knees, 9.7%) of the lateral meniscus. **Conclusions:** The ALL has an attachment to the lateral meniscus in all studied knees, with its most prevalent site at zone 2b. The most frequent types of ALL attachment on the lateral meniscus were full thickness or bipolar. These anatomic insights support targeted anterolateral augmentation and meniscal preservation to optimize clinical results.

## 1. Introduction

The first reference to the anterolateral ligament (ALL) is the 1879 study by the French gynecologist Paul Ferdinand Segond [1]. Whereas Segond [1] described a ‘pearly, fibrous band’ attached to a small avulsed tibial bone fragment, subsequent literature has only rarely mentioned the presence of a ligament connecting the femur with the anterolateral part of the tibia [2,3,4,5]. The inconsistent terminology, ambiguous descriptions, and absence of detailed illustrations have contributed to significant uncertainty regarding this structure’s precise anatomy and function.

Cavaignac and colleagues [4], in a recent historical essay on the ALL, reported that several works from the late 1800s and early 1900s, consistently described a “lateral epicondilo-meniscal ligament”. The debate about the existence, anatomy, and role of the so-called ALL has been one of the main topics of discussion in the orthopedic community over the past years. For instance, Claes and colleagues [5] proposed the anterolateral ligament (ALL) as a consistent, discrete structure with surgical relevance. On the other hand, Musahl and collaborators [6] (including Freddie Fu) urged caution, questioning definitional consistency and routine reconstruction, emphasizing the objective assessment of rotational instability (e.g., pivot-shift), individualized graft choices, and the need for robust clinical outcomes before widespread adoption. With the increased number of published articles [7], the existence of the ALL is no longer the main focus of more recent studies. According to the consensus paper from the ALL Expert Group [8], the ALL is considered a “distinct ligament at the anterolateral side of the human knee”, “deep to the iliotibial band”, with “femoral attachment posterior and proximal to the lateral epicondyle, and a tibial attachment between Gerdy’s tubercle and the fibular head” with a “constant attachment to the lateral meniscus”.

Actually, the mentioned debate has contributed to a progressively better understanding of the ALL, with improved anatomical descriptions of its attachments and dimensions [9,10,11,12]. Furthermore, biomechanical studies suggest that the ALL limits tibial medial (internal) rotation on the femur [12,13]. This knowledge helped develop reconstruction procedures to minimize pivot-shifts following anterior cruciate ligament (ACL) surgery [14].

The lateral meniscus plays a major role in the pivot-shift maneuver as lateral meniscectomy increases the translation and rotation of the tibia relative to the femur, enhancing the pivot-shift phenomenon [15]. Several studies described an attachment of the ALL on the lateral meniscus [10,12,16]. However, to the best of our knowledge, there are only few cadaveric dissection studies that thoroughly analyzed the detailed morphology of this attachment [3,5,17]. Thus, this research aims to describe in detail the attachment of the ALL on the lateral meniscus regarding both the zone and the type of fixation.

## 2. Materials and Methods

The cadavers used in this study were derived from body donation with informed consent, written, and signed by the donator himself (Portuguese Decree-Law No. 274/99). This cadaveric study was approved by the Ethics Committee of the Faculty of Medicine of the University of Porto/RISE Health (366/CEFMUP-RISEHealth/2025). The study was performed on 16 Thiel-embalmed [18] Caucasian cadavers resulting in 32 knees (20 from male cadavers and 12 from females; mean age of 70 years, standard deviation 8 years), one of which was excluded because it had a prosthesis. None of the other knees used had apparently any history of previous surgeries or fractures around the knee that could alter its anatomy. As routine in our unit, precise dissection techniques were employed using appropriate tools to preserve the anatomy of the knee [10,19,20].

A trained knee consultant orthopedic surgeon dissected all knees. A lateral approach of the knee, based on the description of Claes and collaborators [5], was used. Dissection began with a curvilinear incision proximal to the lateral epicondyle and extending distally between the fibular head and Gerdy’s tubercle. After visualization of the iliotibial tract (ITT), a transverse incision was made approximately 6–8 cm proximal to the lateral epicondyle. The ITT was then reflected distally. Careful reflection of the ITT was essential to identify the ALL, especially near the lateral epicondyle where the proximal part of the ALL closely adhered to the ITT. The reflection of the ITT was then taken further distally until its insertion on to Gerdy’s tubercle, and the underlying adipose tissue was removed. When the fibular collateral ligament (FCL) was identified, the leg was medially (internally) rotated with the flexed knee to enhance the visualization of ALL. We observed whether there was dense fibrous tissue that ran superficial to the FCL in the anterolateral part of the knee capsule, which corresponded to the ALL (Figure 1). After identification and marking of the ALL, the knee joints were opened anteriorly by a longitudinal incision on the medial side of the knee capsule, cutting the patellar ligament and the tibial (medial) collateral ligament transversely. To achieve optimal visualization of the lateral meniscus, the joint capsule and the cruciate ligaments were cut.

After careful dissection, the ALL, the lateral meniscus, and the bony landmarks (lateral epicondyle, fibular head, and Gerdy’s tubercule) were identified. A standardized measurement technique was performed using a surgical ruler. The measurements were performed independently by two different observers and presented in millimeters (mm). In the event of a discrepancy (witch only happened in three knees regarding the distance to the tibial articular surface) a third researcher was consulted, and resolution was obtained by consensus prior to analysis. ALL proximal and distal bone insertions were documented, and its length (distance between its proximal and distal attachments points) was measured with a knee flexion of 90°.

The ALL relations with the lateral meniscus were noted. The lateral meniscus can be divided into six zones described from the anterior to the posterior tibial attachment [21] (Figure 2). These include the anterior root (zone 1), the anterolateral zone between the anterior root and the anterior border of the popliteal hiatus (zones 2a and 2b), the popliteal hiatus (zone 3), the posteroinferior popliteomeniscal fascicle (zone 4), the ligamentous zone (zone 5), and the posterior root (zone 6) [21]. ALL attachment on the lateral meniscus was noted according to these zones, and the length of the ALL anteroposterior attachment on the lateral meniscus was measured. The ALL meniscal attachment, regarding the extension of the height of the peripheral border of the lateral meniscus in which ALL had its attachment, was classified into four types: full thickness, superior, inferior, and bipolar (only superior and inferior attachment on the meniscus).

Specifically, regarding the tibial attachment of the ALL, the following measurements were taken: (a) the distance to the tibial articular surface; (b) the distance to Gerdy’s tubercle; and (c) the distance to the fibular head.

Data were collected and stored, anonymously, using Microsoft Excel® (Version 2016; Microsoft Corp, Redmond, Washington, DC, USA), with statistical analysis accomplished with IBM SPSS® 29 (IBM Corp., New York, NY, USA). A descriptive statistical analysis was performed and continuous variables were summarized as median and 25th–75th percentiles (P25–P75), except for age, which presented a normal distribution and was characterized by mean and standard deviation. Categorical variables were reported as frequencies and proportions. The Kruskal–Wallis test was used to compare the anteroposterior length of the ALL attachment on the meniscus across the different types of its attachment on the lateral meniscus. A value of *p* < 0.05 was considered statistically significant.

## 3. Results

A distinct ligamentous structure was identified in all 31 knees at the anterolateral side of the knee joint connecting the femur with the tibia (Figure 1). This structure was easily distinguishable from the thinner joint capsule lying anterior to it. The ALL proximal attachment was located posterosuperiorly and adjacent to the lateral epicondyle of the femur. From its proximal attachment, the ALL coursed in an anterodistal direction to its distal attachment onto the anterolateral aspect of the proximal part of the tibia midway between Gerdy’s tubercle and the fibular head with a median distance of 5.2 mm (P25–P75 4.5–6.0 mm) from the tibial articular surface, a median of 14.3 mm (P25–P75 12.0–16.2 mm) posterior to Gerdy’s tubercle, and a median of 15.2 mm (P25–P75 12.2–16.3 mm) anterior to the fibular head. The median length in neutral rotation and with 90° flexion was 34.3 mm (P25–P75 31.3–35.5 mm).

The ALL had an attachment to the lateral meniscus in all studied knees. The ALL was attached in zone 2b of the lateral meniscus in 97% of cases and in zone 2a in 3% of cases. The median anteroposterior length of the ALL attachment on the lateral meniscus was 6 mm (P25–P75 5–7 mm). The patterns of attachment were 41.9% full thickness (Figure 3), 35.5% bipolar (Figure 4), 12.9% superior-only (Figure 5), and 9.7% inferior-only (Figure 6) (Table 1). No statistically significant association was found between the anteroposterior length of the ALL attachment on the meniscus and the type of its attachment on the lateral meniscus (*p* = 0.222).

## 4. Discussion

One of the most important finding of this study is that the ALL was observed in 100% of the studied knees, and there was always an attachment of the ALL on the lateral meniscus.

Clinical and anatomical studies about the ALL have increased significantly in recent years. The prevalence of ALL in the literature is quite variable, ranging from 0% to 100% [5,10,12,16,17,22,23,24,25]. One explanation for this is a possible interference of the dissection technique with proper ALL identification and the use of embalmed cadavers, which may hinder the dissection.

In this work, using a dissection technique based on that described by Claes and colleagues [5], the ALL was found in 100% of the knees. As previously described, its proximal attachment was closely related to the FCL femoral attachment site [20]. We found that the ALL proximal attachment was located posterosuperiorly and adjacent to the lateral epicondyle of the femur. Regarding the tibial attachment of the ALL, it has been consistently described [20]. In this study, we found it halfway between the head of the fibula and Gerdy’s tubercle, which is also in agreement with early published descriptions [5,10,12]. Concerning the distance of the tibial attachment to the tibial articular surface, we found a median distance of 5.2 mm (P25–P75 4.5–6.0 mm), a value that was slightly higher than that observed (2.1 ± 0.6 mm) in our previous study [10]. The length of the ligament measured in this work is approximately similar to other studies in the literature [3,5,10,12,22].

In the present study, an anatomical relation between the ALL and the lateral meniscus was verified in all knees, confirming the presence of an ALL meniscal insertion. Indeed, multiple cadaveric dissection studies support the existence of a meniscal attachment of the ALL [3,5,10,12,17,22,25,26]. A recent systematic review with meta-analysis [16] showed that the ALL was attached to the lateral meniscus periphery in 97% of studies. Among existing studies, Vincent and collaborators [3] located this meniscal attachment near the junction of its anterior and middle thirds, Helito and colleagues [17] in the transition between the anterior horn and the meniscus body, whereas Claes and colleagues [5] located it in the middle third of the meniscal body of the lateral meniscus. No cadaver study described the precise type of connection of the ALL to the lateral meniscus regarding both its anteroposterior and cephalocaudal extent. Actually, this detail might not be obvious macroscopically and can be compromised by the dissection technique. To overcome this difficulty, a standard protocol was herein used for the dissection technique. This work illustrates a stable location of the ALL attachment on the 2b zone of the lateral meniscus, which is anterior to the popliteal hiatus, in line with the most frequent zone reported by Helito and colleagues [17]. Classically, four types of meniscal attachment can be identified through magnetic resonance imaging: complete, central, bipolar, or inferior-only [27]. In our study, we had interesting results showing that almost 80% of meniscal attachments were full-thickness (41.9%) or bipolar (35.5%). The clinical significance of this classification is not yet fully known. Different attachment patterns may vary in mechanical strength, potentially influencing injury patterns observed in the ALL and the lateral meniscus in injured knees. Despite this, no statistical association was found between the anteroposterior length of the insertion of the ALL and the type of attachment on the lateral meniscus.

Some biomechanical studies have suggested a role for the anterolateral portion of the knee joint capsule in limiting tibial medial (internal) rotation on the femur [28], which has led to the proposed reconstruction procedures to limit pivot-shift following ACL surgery [14]. At the time of the reconstruction, in addition to the bony references of the femur and tibia, combined with the knowledge about ALL isometric pattern [29], the meniscal attachment may also be important. Furthermore, ALL injury may also be associated with peripheral disinsertion of the lateral meniscus. The lateral meniscus and its relation with ALL motion was tackled by Caterine and collaborators in their work [9], and Helito and colleagues [17,30] in the discussions of their anatomical studies, raised the possibility of the existence of these combined injuries. An epidemiological study on ACL injuries concluded that lateral meniscus injuries usually occurred at the time of an ACL injury and not over time, as occurs for the medial meniscus [31], and Claes and colleagues [32] showed a high rate of ALL abnormalities in patients with an ACL injury. One of the reasons for this combined lesion may be their close anatomical relationship with the ALL. In this study, we had difficulty distinguishing an evident separation between the meniscus fibers and the ligament attachment. Thus, it is possible that the ALL may participate in the genesis of lateral meniscus tears, especially peripheral rim detachment tears. This anatomic relationship supports the concept that anterolateral restraint and lateral meniscal mechanics are interdependent. These findings have direct clinical implications. First, they help refine patient selection for anterolateral augmentation during ACL reconstruction: patients presenting with a high-grade pivot-shift, imaging or intra-operative signs of anterolateral complex injury, or concomitant lateral meniscus lesions (including posterior horn/root involvement) may particularly benefit from selective augmentation alongside meniscal-sparing repair and protection strategies [33]. Second, the delineated landmarks and the ALL–meniscus line of force can improve intra-operative reproducibility, guiding graft orientation, tunnel position, and tensioning at appropriate flexion/rotation to restore rotational control without over-constraint of the lateral compartment. Finally, recognizing variant patterns of ALL–meniscus connection could provide a pathophysiological framework for combined injuries, informing both pre-operative planning (imaging interpretation, risk stratification) and post-operative expectations. Prospective kinematic and outcome studies should test these hypotheses and quantify their impact on clinical results.

Although our study has produced important clinical anatomical findings, which we hope can influence surgical practice, it also has some limitations that are generally observed in studies that are performed with cadavers. Nevertheless, we attempted to minimize these limitations. The main limitation is related to the changes in the volume and trophicity of tissues due to death and fixation techniques. For instance, embalming techniques may alter tissue properties (e.g., hydration, pliability and color contrast), which can reduce the capability to accurately identify thin capsuloligamentous structures such as the ALL. This could have influenced both the identification and measurement in our cadaveric dissections. We tried to minimize this by using bony references. Furthermore, we measured some reference parameters, and the results obtained were similar to the data previously reported in the literature, which unequivocally provides robustness to our results. Finally, due to limited availability and associated cost, the number of cadaver specimens used was relatively low.

## 5. Conclusions

With a standard and careful dissection technique, in our study, the ALL was found in all cases. The attachment site on the lateral meniscus was almost always the same, and the pattern of attachment was variable, being most frequently full thickness or bipolar. This study corroborates the growing evidence that the discussion must move from the question of the existence of the ALL to the assessment of its morphology, its anatomical relations, and its biomechanical and clinical significance. These data outline imaging targets and technique cues that can standardize in vivo identification of ALL–meniscus coupling and optimize graft orientation/tensioning in ACL with or without anterolateral reconstruction.

## Figures and Tables

**Figure 1 diagnostics-15-02987-f001:**
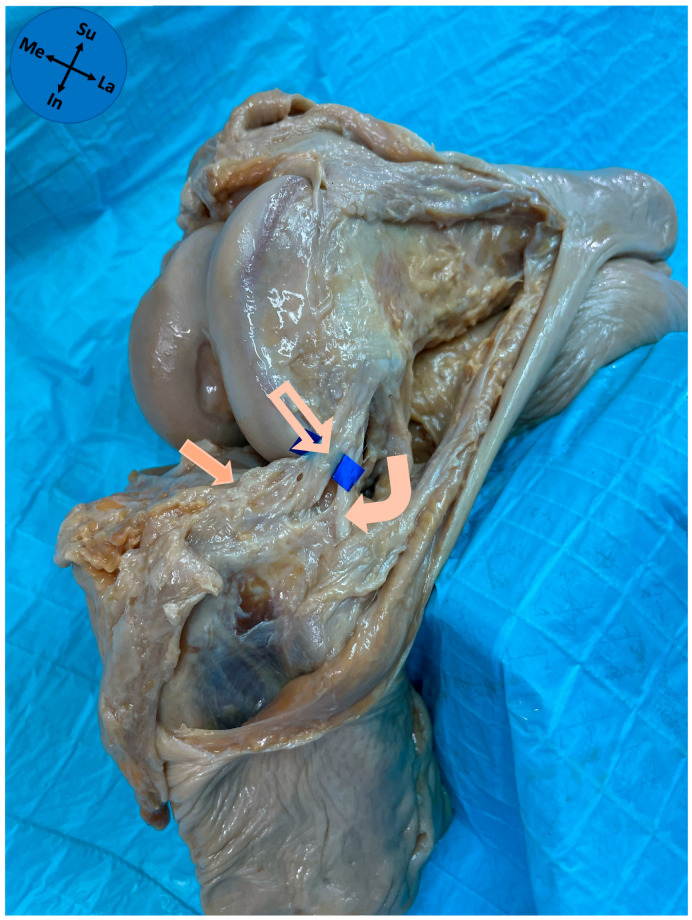
Anterolateral view of an anatomical dissection of a left knee. Solid arrow—lateral meniscus; Outline arrow—anterolateral ligament; Curved arrow—fibular collateral ligament; Me—medial; La—lateral; In—inferior; Su—superior.

**Figure 2 diagnostics-15-02987-f002:**
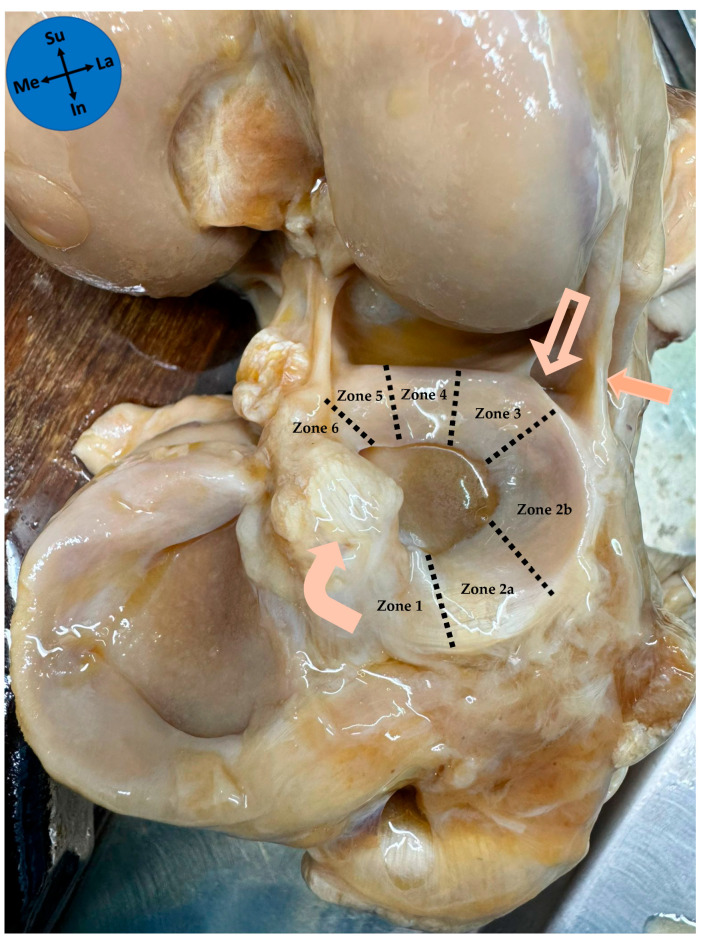
Zones of the lateral meniscus of a left knee. Solid arrow—anterolateral ligament; Outline arrow—popliteal tendon; Curved arrow—anterior cruciate ligament; Me—medial; La—lateral; Su—superior; In—inferior.

**Figure 3 diagnostics-15-02987-f003:**
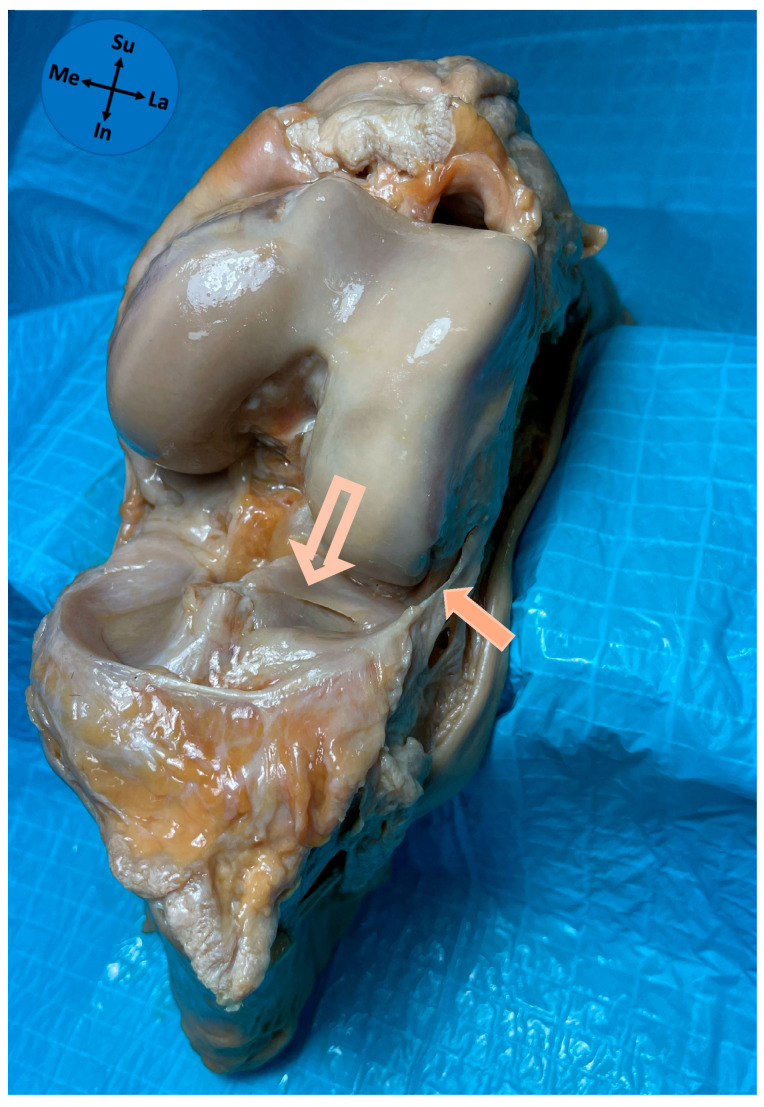
Full thickness type of ALL attachment on the lateral meniscus. Solid arrow—anterolateral ligament; Outline arrow—lateral meniscus; Me—medial; La—lateral; Su—superior; In—inferior.

**Figure 4 diagnostics-15-02987-f004:**
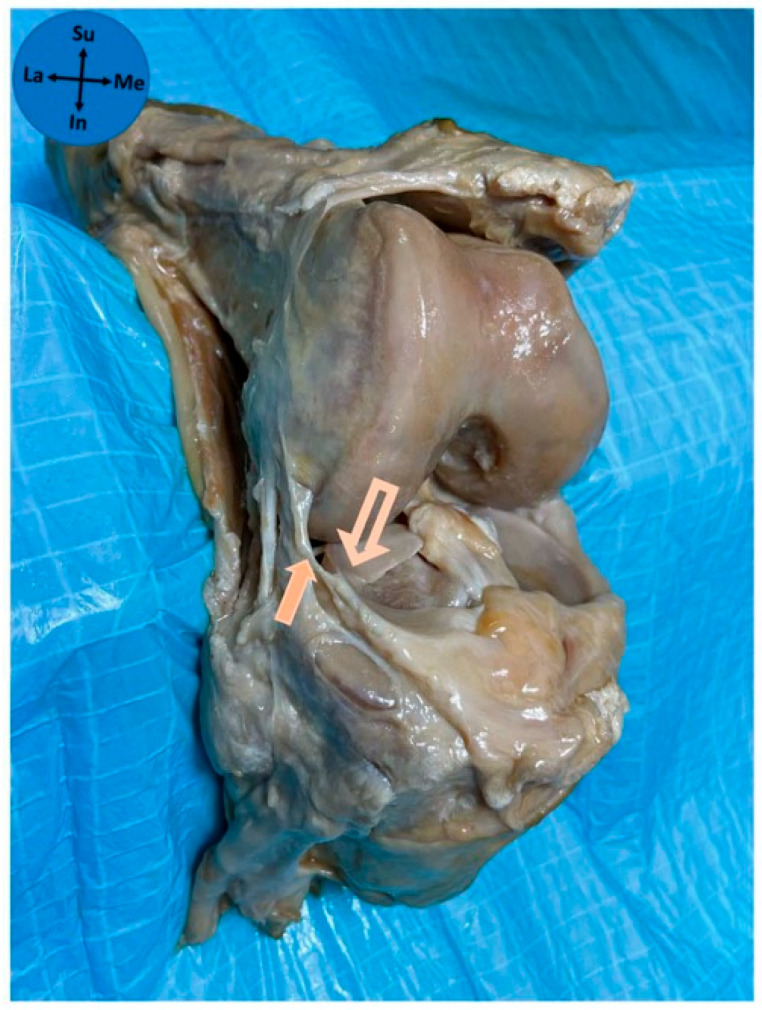
Bipolar type of ALL attachment on the lateral meniscus. Solid arrow—anterolateral ligament; Outline arrow—lateral meniscus; Me—medial; La—lateral; Su—superior; In—inferior.

**Figure 5 diagnostics-15-02987-f005:**
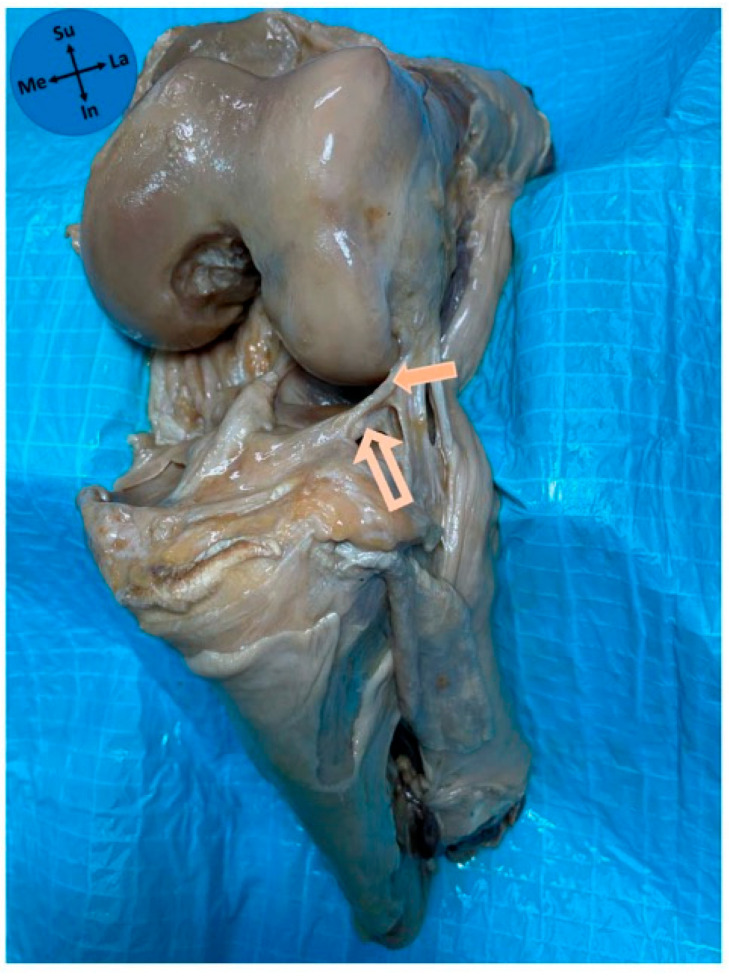
Superior type of ALL attachment on the lateral meniscus. Solid arrow—anterolateral ligament; Outline arrow—lateral meniscus; Me—medial; La—lateral; Su—superior; In—inferior.

**Figure 6 diagnostics-15-02987-f006:**
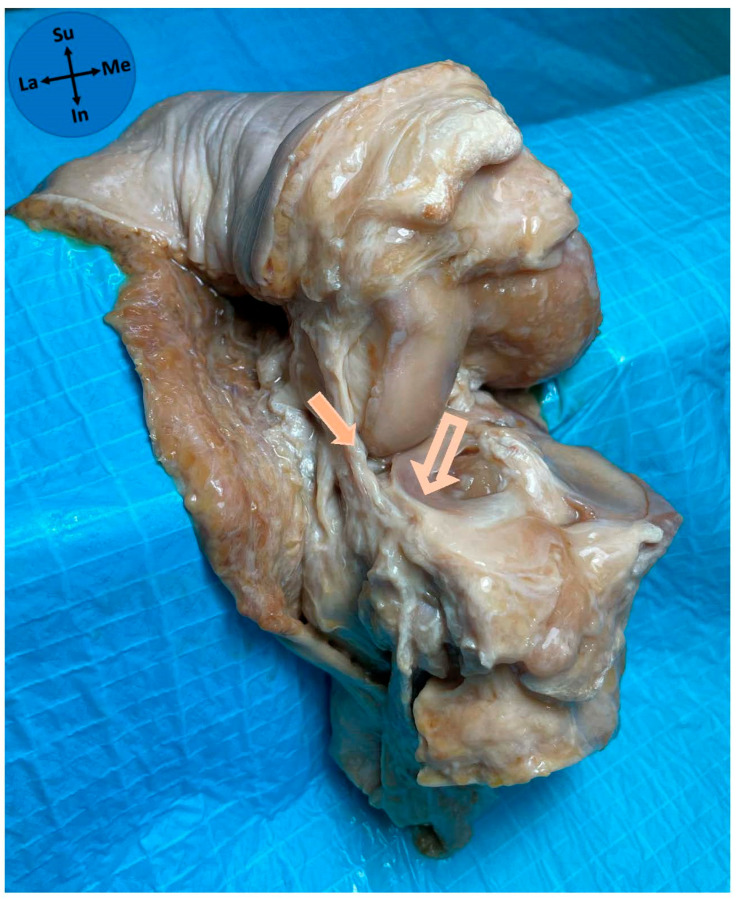
Inferior type of ALL attachment on the lateral meniscus. Solid arrow—anterolateral ligament; Outline arrow—lateral meniscus; Me—medial; La—lateral; Su—superior; In—inferior.

**Table 1 diagnostics-15-02987-t001:** Characterization of ALL attachments to the lateral meniscus (*n* = 31 knees).

Meniscal Zone of Attachment	*n* (%)
Zone 2a	1 (3.2)
Zone 2b	30 (96.8)
**Pattern of meniscal attachment**	***n* (%)**
Full thickness	13 (41.9)
Bipolar	11 (35.5)
Superior-only	4 (12.9)
Inferior-only	3 (9.7)

ALL—anterolateral ligament of the knee.

## Data Availability

The raw data supporting the conclusions of this article will be made available by the authors on request.

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
