# Peer review of "Anatomical Insights into the Lateral Meniscus and Anterolateral Ligament: A Cadaveric Study"

_diagnostics, 2025, doi:10.3390/diagnostics15232987_

Round 1
Reviewer 1 Report
Comments and Suggestions for Authors
Dear Authors
congratulations on a fine paper
and on a not anymore controversil paper
why do nt you mention any ref to the late Freddy Fu who was in constant " conflict " with steven Claes
see some suggestions / comments on balloons in your paper
the language is ok
title and tables are ok
i feel like the images and annotations are not " clear" and may be improved . i wonder if just a few small arrows might not add to clarity
refs are ok ... i miss FFu papers

Author Response
good morning
A point-by-point response to the reviewer was done.
|
Response to Reviewer 1 Comments
|
||||
|
1. Summary |
|
|
||
|
We wish to thank the Reviewer for carefully reading the manuscript and for the very useful comments and suggestions. Please find the detailed responses below and the corresponding revisions/corrections highlighted/in track changes in the re-submitted files.
|
||||
|
2. Point-by-point response to Comments and Suggestions for Authors |
||||
|
Comments 1: Why don´t you mention any ref to the late Freddy Fu who was in constant " conflict " with steven Claes
|
||||
|
Response 1: Thank you for pointing this out. We appreciate the suggestion. Recognizing the importance of Professor Freddie H. Fu in this field, we will add key references to his work and that of his collaborators to better contextualize the state of the art and the discussion of our results. (page number 2, 2º paragraph, line 53-58)
|
||||
|
Comments 2: I feel like the images and annotations are not " clear" and may be improved. I wonder if just a few small arrows might not add to clarity.
|
||||
|
Response 2: We agree. We have, accordingly, modified the images to emphasize this point with different arrows. (Page 5-10)
|
||||
Best regards
Reviewer 2 Report
Comments and Suggestions for Authors
This is a well-conducted anatomical cadaveric study addressing the relation between the anterolateral ligament (ALL) and the lateral meniscus. The paper is clearly written, well-structured, and supported by detailed dissection methodology and relevant references. The topic is timely and relevant, as the role of the ALL and its meniscal attachment remains an area of active research in orthopedic anatomy and surgery. The findings are consistent with existing literature and add precision regarding the type and prevalence of ALL-meniscal attachment.
Overall, this work provides valuable anatomical insights that can support future biomechanical and clinical investigations. Nevertheless, there are a few points that require clarification or improvement before publication.
Abstract:
The abstract is well written and concise. However, it may benefit from a brief statement about the clinical implications of the findings, beyond the descriptive anatomy.
Introduction:
The background is comprehensive, with historical and contemporary references. Consider shortening some historical details to focus more on the rationale for this cadaveric study and the knowledge gap (only one previous precise report of meniscal insertion).
Materials and Methods:
- The methodology is clear and reproducible, with appropriate references to standardized dissection techniques.
-
Please clarify whether interobserver agreement was assessed quantitatively (e.g., kappa statistics) rather than only resolving discrepancies by consensus.
-
Statistical analysis is adequate, though limited to descriptive and non-parametric testing. This is acceptable given the anatomical focus.
Results:
-
Results are presented clearly with tables and figures.
-
Figures 3–6 are helpful but could benefit from higher resolution and possibly arrows or markers to improve readability.
-
Table 1 is concise and appropriate.
Discussion:
-
The discussion is thorough, with strong integration of findings into existing literature.
-
Some parts could be shortened to avoid redundancy (e.g. repetition of attachment sites compared with prior studies).
-
The authors should more explicitly discuss clinical significance: How could different attachment patterns influence surgical reconstruction, risk of meniscal tears, or pivot-shift instability?
-
The limitations section is appropriate, but it could further highlight that embalming techniques may alter tissue properties, potentially affecting visibility of thin structures like the ALL.
Conclusion:
-
Strong conclusion that shifts the debate from the existence of the ALL to its morphology and biomechanics.
-
Consider adding one sentence on how these findings may guide future imaging studies or surgical reconstruction techniques.
Language and Style:
-
Overall English is clear and professional, but minor grammatical polishing would enhance readability (e.g. line 29: “the ALL has an attachment in zone 2b” -> “the ALL was attached in zone 2b”).
-
The manuscript would benefit from careful proofreading to eliminate small redundancies.
Strengths:
-
Large sample size for an anatomical cadaveric study (31 knees),
-
Standardized dissection protocol,
-
Clear identification and classification of ALL-meniscus attachment patterns,
-
Comprehensive integration with prior anatomical and biomechanical literature.
Weaknesses:
-
Limited statistical depth (though appropriate for anatomical work),
-
Some redundancy in the discussion,
-
Figures could be more illustrative with markings,
-
Clinical relevance of different attachment patterns not fully elaborated.
Recommendation - Major Revision
The manuscript is scientifically solid and offers valuable anatomical insights with clear potential for publication. Nevertheless, several issues must be addressed before it can be accepted:
-
Discussion: At present, the discussion is overly detailed and somewhat repetitive. It should be streamlined, with greater focus on the clinical implications of the findings. The authors are encouraged to highlight how their results may impact surgical decision-making, reconstruction techniques, or the understanding of combined ALL–meniscus injuries.
-
Figures (3–6): While informative, these figures are of relatively low resolution. Their clarity would be greatly enhanced by higher-quality images with arrows or other markers to guide the reader’s attention.
-
Clinical relevance: The manuscript briefly mentions the possible significance of different ALL–meniscus attachment patterns, but this remains underdeveloped. A deeper discussion on how these patterns might influence biomechanics, injury mechanisms, or outcomes of ACL/ALL reconstructions would strengthen the paper considerably.
-
Language and style: Although the English is generally clear, the manuscript would benefit from professional editing to improve fluency, remove redundancies, and align with the polished academic style.
In summary, the study provides a robust anatomical contribution, but revisions are necessary to improve readability, strengthen clinical interpretation, and ensure the highest scientific quality.
Comments on the Quality of English LanguageThe English language is generally clear and understandable, but the manuscript would benefit from careful professional editing. Several sentences are repetitive or stylistically heavy, and minor grammatical adjustments are needed to improve fluency. Polishing the text will ensure that the presentation meets the high academic standards expected in Diagnostics.
Author Response
Good morning
A point-by-point response to the reviewer was uploaded.
|
Response to Reviewer 2 Comments
|
||
|
1. Summary |
|
|
|
We wish to thank the Reviewer for carefully reading the manuscript and for the very useful comments and suggestions. Please find the detailed responses below and the corresponding revisions/corrections highlighted/in track changes in the re-submitted files.
|
||
|
2. Point-by-point response to Comments and Suggestions for Authors
|
||
|
Comments 1: The abstract is well written and concise. However, it may benefit from a brief statement about the clinical implications of the findings, beyond the descriptive anatomy. |
||
|
Response 1: Thank you for pointing this out. We agree with this comment. These anatomic relationships have direct clinical relevance, helping refine indications for anterolateral augmentation during ACL reconstruction, guide graft/tunnel positioning, and support strategies to protect the lateral meniscus, with the goal of improving control of rotational instability while avoiding over-constraint. (Page 1, Abstract, Line 36-37)
|
||
|
Comments 2: The background is comprehensive, with historical and contemporary references. Consider shortening some historical details to focus more on the rationale for this cadaveric study and the knowledge gap (only one previous precise report of meniscal insertion). |
||
|
Response 2: We agree. We have condensed the historical background to a few essential sentences and shifted the emphasis to the study rationale and knowledge gap. We removed non-essential historical detail, retained only the key context, and added explicit statements outlining why a cadaveric approach is required and that there is only one previous precise report of the meniscal insertion on this topic. |
||
|
Comments 3: Please clarify whether interobserver agreement was assessed quantitatively (e.g., kappa statistics) rather than only resolving discrepancies by consensus. |
||
|
Response 3: Thank you. We did not quantify interobserver agreement in this study. Still, we clarified this issue in the text (Pag 3, Line 110-114). |
||
|
|
||
Comments 4: Figures 3–6 are helpful but could benefit from higher resolution and possibly arrows or markers to improve readability.
Response 4: Thank you for the suggestion. We have added clear arrows/markers to identify the key anatomical structures. We also standardized arrow style and thickness across panels and updated the figure legends to reflect these annotations, improving readability and interpretability.
Comments 5: At present, the discussion is overly detailed and somewhat repetitive. It should be streamlined, with greater focus on the clinical implications of the findings. The authors are encouraged to highlight how their results may impact surgical decision-making, reconstruction techniques, or the understanding of combined ALL–meniscus injuries.
Response 5: Thank you for the constructive comment. We streamlined the Discussion to emphasize clinical implications, explicitly detailing how our anatomic findings may influence surgical decision-making, reconstruction technique, and the understanding of combined ALL–lateral meniscus injuries. The revised text appears in green in the Discussion (Pag 12-13, Line 430-444).
Comments 6: The limitations section is appropriate, but it could further highlight that embalming techniques may alter tissue properties, potentially affecting visibility of thin structures like the ALL.
Response 6: Thank you for the helpful suggestion. We have added an explicit limitation noting that embalming can alter soft-tissue properties (e.g., hydration, elasticity, and visual contrast), which may reduce the visibility of thin capsuloligamentous layers such as the ALL and potentially influence identification and measured dimensions. This wording has been incorporated into the limitations section to contextualize our findings (Pag 13, Line 449-452).
Comments 7: In the conclusion consider adding one sentence on how these findings may guide future imaging studies or surgical reconstruction techniques.
Response 7: Thank you for the suggestion. We have added a concise sentence to the Conclusions highlighting implications for imaging and reconstruction techniques (Pag 13, Line 464-466).
|
3. Additional clarifications |
|
The suggestions/comments were all taken to account and modifications were made to improve the paper. |
Best regards
Round 2
Reviewer 2 Report
Comments and Suggestions for Authors
The revised version of your manuscript shows substantial improvement and fully addresses all comments from the previous review. The Introduction is now concise and well-focused on the study rationale and existing knowledge gap. The Methods section is clearly described and includes appropriate ethical and statistical details. Figures have been improved with clear annotations, and the Results are presented in a structured and transparent manner.
The Discussion has been effectively streamlined and now emphasizes the clinical relevance of the ALL–lateral meniscus relationship, providing meaningful implications for anterolateral reconstruction and meniscal preservation. The limitations section appropriately acknowledges the potential influence of embalming on soft-tissue visualization.
Overall, this is a well-executed and carefully revised manuscript that contributes valuable anatomical and clinical insights into the anterolateral complex of the knee. Only very minor typographical and stylistic corrections remain, which can be handled during copy-editing.